# Cryo-EM Structures of Two Bacteriophage Portal Proteins Provide Insights for Antimicrobial Phage Engineering

**DOI:** 10.3390/v13122532

**Published:** 2021-12-16

**Authors:** Abid Javed, Hugo Villanueva, Shadikejiang Shataer, Sara Vasciaveo, Renos Savva, Elena V. Orlova

**Affiliations:** 1Department of Biological Sciences, Institute of Structural and Molecular Biology, Birkbeck College, Malet Street, London WC1E 7HX, UK; abid_javed@ymail.com (A.J.); hugo.villanueva92@gmail.com (H.V.); sadik08@gmail.com (S.S.); sara.vasciaveo.20@ucl.ac.uk (S.V.); r.savva@mail.cryst.bbk.ac.uk (R.S.); 2Astbury Centre for Structural Molecular Biology, Faculty of Biological Sciences, University of Leeds, Leeds LS2 9JT, UK; 3Stevenage Bioscience Catalyst, Gunnels Wood Road, Stevenage SG1 2FX, UK

**Keywords:** bacteriophage, GA1, phiCPV4, portal protein, cryo electron microscopy, structural analysis, clip domain

## Abstract

Widespread antibiotic resistance has returned attention to bacteriophages as a means of managing bacterial pathogenesis. Synthetic biology approaches to engineer phages have demonstrated genomic editing to broaden natural host ranges, or to optimise microbicidal action. Gram positive pathogens cause serious pastoral animal and human infections that are especially lethal in newborns. Such pathogens are targeted by the obligate lytic phages of the *Salasmaviridae* and *Guelinviridae* families. These phages have relatively small ~20 kb linear protein-capped genomes and their compact organisation, relatively few structural elements, and broad host range, are appealing from a phage-engineering standpoint. In this study, we focus on portal proteins, which are core elements for the assembly of such tailed phages. The structures of dodecameric portal complexes from *Salasmaviridae* phage GA1, which targets *Bacillus pumilus*, and *Guelinviridae* phage phiCPV4 that infects *Clostridium perfringens*, were determined at resolutions of 3.3 Å and 2.9 Å, respectively. Both are found to closely resemble the related phi29 portal protein fold. However, the portal protein of phiCPV4 exhibits interesting differences in the clip domain. These structures provide new insights on structural diversity in *Caudovirales* portal proteins and will be essential for considerations in phage structural engineering.

## 1. Introduction

In an era of widespread antibiotic resistance, there is a need to explore alternative treatments to treat bacterial pathogenesis, such as phage therapy, in which bacteriophages are administered as therapeutic antimicrobials. With the maturation of synthetic biology and integrated structural biology, bacteriophages can now be engineered both at the level of the genome (payload) and structurally to offer design-based means to circumvent phage-resistance in bacteria [1]. While sequence-informed gene substitutions between related phages have been successful demonstrations of the utility of this as an approach, it is likely that some sequence-related structural components may prove incompatible between any two phages [2]. Bacteriophage engineering was focused so far mainly on Gram-negative pathogens, which leaves a substantial burden of Gram-positive infections outside the scope of this useful therapeutic alternative to antibiotics. The smallest known tailed bacteriophages are the *Salasmaviridae, Guelinviridae* and *Rountreeviridae*, and represent families of bacterial viruses in the order *Caudovirales* (tailed dsDNA phages) characterised by their short linear DNA genomes (the observed genome length range at the time of writing is c. 16.5–27.5 kb). To target a bacterial cell, these phages bind to host-specific types of teichoic acids by specialised appendages and then infect by hydrolysing the peptidoglycan crosslinks via the tail tip rather than using a specific bacterial receptor to inject the viral genome [3]. The representative species of these phage families share an unusual DNA replication mechanism primed by terminal proteins covalently bound to the linear DNA genome termini [4,5]. The *Salasmaviridae, Guelinviridae and Rountreeviridae* phages are feasible candidates for phage engineering due to their small genome size, morphological resemblance, and the range of important human and animal pathogens they target (*Bacillus, Clostridium*, *Staphylococcus*, *Streptococcus*, *Enterococcus*). The phage phi29 is the most studied phage of the *Salasmaviridae* family [6,7,8,9]. The GA1 phage of the *Salasmaviridae* family targets the pathogen *Bacillus pumilus*, causative of serious infant infections. The phiCPV4 phage belongs to the *Guelinviridae* family and infects *Clostridium perfringens*, a pathogenic scourge of the poultry industry, which causes foodborne illnesses in humans.

The Portal Protein (PP) complex is a central component of *Caudovirales* virions in the process of self-assembly of phages. The described PPs in all phages studied to date are characterised by the conservation of major motifs and an overall structural fold at the level of the monomer (Figure 1), despite low sequence similarity in PP gene products and a wide variation of monomeric molecular mass [6,10,11,12]. The PP has multiple functions: (1) as a nucleation point for capsid formation [9]; (2) as part of the motor for loading the genome into the capsid [13]; (3) as a head-full sensor [6]; and (4) as a conduit for genome translocation [14]. Finally, the PP serves as a linker or base for the tail complex [15]. Once the genome is fully loaded, the PP in tandem with terminal proteins are purported to serve as a base for lower collar neck protein (gp11) oligomerisation, from which the tail is assembled [8]. In complexes with adaptor proteins, the PP acts as a gatekeeper by preventing genomic leakage from the capsid and to open the channel for injecting the genome into a host bacterium via the tail [14].

Analysis of portal protein phylogeny (Figure 2). indicates that *Salasmaviridae* (Figure 2, in blue) infecting *Bacillus* spp., and *Streptococcus* spp., are more closely related to *Guelinviridae* (Figure 2, in yellow) than the more distant *Rountreeviridae* (Figure 2, in red) infecting *Staphylococcus* spp., or *Enterococcus* spp. Historically, phiCPV4, GA1, phi29 and P68 were classified under the sub-family of *Picovirinae* of *Podoviridae*, due to close morphological relationships. Changes in classification in 2021 by the ICTV based on genomic features [16], reclassified these phages in the orders of *Guelinviridae, Salasmaviridae and Rountreeviridae* [17]. Nonetheless, morphological similarities remain relevant for the aims of this study. *Salasmaviridae,* which infect *Bacillus* and *Streptococcus* species, cluster together with GA1.

Homologues of phi29, GA1 and phiCPV4 portal proteins were found using PSI-BLAST [18]. The GA1 PP (gp10) has 52.4% sequence identity with the phi29 PP (gp10). *Guelinviridae* infecting *Clostridium* species also cluster together (Figure 2, in yellow) with the phiCPV4 PP (gp17) sharing 28.7% sequence identity with phi29 gp10 (Appendix A). 

Portal proteins are mechanistically and structurally the best studied proteins in *Caudovirales*, and due to their central role in phage morphology and function, could be engineered to create a chimeric portal. Such a chimera could in principle attach the capsid of one phage to the tail of another. Therefore, structural insights related to the interaction of the portal protein with the tail could inform portal engineering approaches aimed at tail switching to redirect a shared capsid and engineered genome to selected microbial targets. Tail switching has been demonstrated in phages targeting gram-negative microbes, like the archetypal *Escherichia* phage T7 [2].

All portal protein primary structures are characterised by a common fold comprising four conserved domains: The capsid-facing C-terminal crown, the interior wing and stem, and the tail-facing clip domains (Figure 1B) [12]. All portal protein structures described to date are dodecameric oligomers exhibiting a ‘turbine shaped’ architecture that suggests a common origin in both *Caudovirales* and *Herpesviridae*. Recent CryoEM structures of phage particles phi29, the *Salasmaviridae* phage of the *Salasvirus* genus (host species, *B. subtilis*), and P68, a *Rountreeviridae* phage of the *Rosenblumvirus* (host species, *S. aureus*) in different states, have highlighted portal protein conformational variations reflecting their dynamics and their functions as packaging motors [8,19].

In the present study, we analysed the portal protein structures of two phages: GA1, targeting *Bacillus pumilus*, and phiCPV4, targeting *Clostridium perfringens,* using cryoEM with computational methods to provide insights that could support engineering designed tail switching aims. The structures of the GA1 and phiCPV4 PPs were obtained at near atomic resolution of 3.3 Å and 2.9 Å, respectively. Comparison of these structures to the portal proteins of phi29 and other phages demonstrates that the GA1 and phiCPV4 PPs closely resemble the fold of the phi29 PP. An interesting discovery is that the phiCPV4 PP exhibits a significant structural distinction within the clip domain compared to the phi29 and GA1 PPs.

## 2. Materials and Methods

### 2.1. Cloning, Expression, and Purification

Synthetic DNA was sourced from Integrated DNA Technologies (IDT), and DNA modifying enzymes and *E. coli* strains were sourced from New England BioLabs (NEB), unless otherwise stated. Bacteriophage GA1 (DSM-5548) was propagated on *Bacillus pumilus* (DSM-5549). The GA1 portal protein was amplified to encode a Strep-tag II/TEV cleavage sequence at the N-terminus with primers AAACCTGTACTTCCAGTCCGTGTTAGAAGATGGTTTTTCTTATAAGACC and GAATGGGCAACACAGTTTGATGATCCGGCTGCTAACAAAG and joined via Gibson Assembly to a pRSET-A (Invitrogen) backbone. A Genestring (IDT) encoding the phiCPV4 portal protein sequence with overlapping overhangs to the vector was sub-cloned into a NheI/HindIII (NEB) linearised pRSET-A (Invitrogen) backbone via Gibson Assembly to encode a hexahistidine tag at the N-terminus.

The *E. coli* strain NEB-5α was used for cloning while the *E. coli* strain T7 Express *lysY/I^q^* was used for protein expression. Recombinant expression was performed overnight at 37 °C in LB-Miller media in 12 × 500 mL Erlenmeyer flasks. Pure GA1 gp10 was obtained via a 2-step purification using a StrepTrap™ HP 1 mL column and 120 mL Superdex HiLoad 200 pg^®^ column (20 mM Tris pH8, 150 mM NaCl, 0.1 mM PMSF). The Strep-tag II was cleaved overnight using TEV protease (3:1 ratio). Pure phiCPV4 gp17 was obtained similarly, with the same buffer but using a HisTrap HP 1 mL column and eluted with a gradient of imidazole (50 mM to 500 mM).

### 2.2. Bioinformatics

Clustal Omega was used with default settings [20]. All residues are coloured using %Equivalent similarity scheme (default threshold 0.7) as implemented in ESPript 3.0 [21]. After removing hits to genomes other than phages, the portal proteins were aligned using the Clustal Omega algorithm after which a maximum likelihood tree was inferred via the Jones-Taylor-Thornton Matrix using the MEGA (Appendix A) [22,23].

### 2.3. Grid Preparation and Data Collection

Purified naïve complexes of GA1 gp10 and phiCPV4 gp17 (at ~0.05 mg/mL each) were deposited to lacey carbon grids (EM Sciences, Radnor Corporate Center, Radnor, PA, USA). Grids with sample were vitrified using a Vitrobot Mark IV (Thermo Fisher Scientific Inc., Hillsboro, OR, USA) at 100% humidity and 8 °C. Purified sample aliquots (3 μL) were applied to grids, incubated for 30 s, blotted for 3.5 s (blot force 6) and then plunge frozen in [liquid nitrogen cooled] liquid ethane. Grids were clipped and transferred to the microscope for data collection. Individual datasets were collected using EPU software (Thermo Fisher Scientific Inc., Hillsboro, OR, USA) on a Titan Krios (Thermo Fisher Scientific Inc., Hillsboro, OR, USA) electron microscope, equipped with a K2 (Gatan, Inc., Pleasanton, CA, USA) direct electron detector. 

Overall, 2249 movies (GA1) and 2903 movies (phiCPV4) were recorded, each movie containing 50 frames, with a dose of 0.98 e^-^/Å^2^ (Total dose of 39.2 e/Å^2^) and 0.985 e^−^/Å^2^ per frame (total dose of 39.4) for GA1 and PhiCPV4 respectively. All data was collected with a pixel size of 1.05 Å/pixel (Appendix A).

### 2.4. Image Processing 

Movies were aligned using MotionCor2 [24]. Aligned micrographs were used to assess CTF parameters using CTFFIND4 (Grigorieff Lab., U. Mass, Worcester, MA, USA) [25]. Micrographs were then manually assessed for CTF quality, based on presence of high-resolution Thon rings as well as absence of astigmatism or ice-quality related defects. A selection of micrographs was then used to process further using RELION 3.0 (MRC Laboratory of Molecular Biology, Cambridge, UK) [25]. 

Particles were initially picked manually in RELION to generate 3D templates for subsequent auto-picking using Gautomatch [26]. Fifty randomly selected micrographs (at a range of defoci) for each dataset were used for manual picking in RELION. Picked particle images were extracted in boxes of size 300 × 300 pixels and used to generate an initial low resolution 3D map using ab-initio 3D in RELION. The 15 Å map was used to make 2D projections in a range of directions (along the big circle) from the top view to the bottom view (14 projections). These projections were used as templates for auto-picking using Gautomatch. For the GA1 and PhiCPV4 datasets, 50,466 and 89,560 particle images of side views were selected correspondingly for each sample. Coordinates of the particle images were imported to RELION 3.0 and particle images were extracted from micrographs using RELION.

Extracted particle images were subjected to 2D classification to clean the dataset such that featureless images were discarded. To assess symmetry of portal complexes, 2D classes that showed end views of portal complexes were used as the basis for symmetry analysis in the data set. First, particles in the end-view 2D classes were sub-classified to assess variations in rotational symmetry, a common occurrence in naïve preparation for portal proteins [27,28]. This assessment of rotational symmetry was done by calculations of the rotational power spectra profiles using IMAGIC_5 [29]. With phiCPV4 gp17, most end view classes showed a 12-mer portal complex whereas GA1 gp10 showed 44% of the particles belonging to a 12-mer complex whilst the remaining belonged to a 13-mer and unclear oligomers. 

Both datasets were subsequently subjected to initial asymmetrical 3D reconstruction using reconstructions with symmetry C1 generated from particle images comprising classes corresponding to side-views and particle images composing a few 2D classes showing the 12-fold symmetry end views; (in RELION), That was followed by 3D classification (Appendix A). Some classes clearly revealed 12-mer structures, while others were distorted and had unclear symmetry. For further refinement, particle images that comprised 3D classes with 12-mer portal complexes were selected and subjected to the following rounds of 3D classification and refinement. In the last steps of the 3D refinement, we used primarily particle images representing side views of the portal complexes with secondary structure features: 41,232 in the GA1 dataset and 65,791 side view images in the PhiCPV4 dataset (Appendix A).

The resulting 3D refined maps were then subjected to 3D classification in two ways: For GA1, 3D classification was run with alignments with the refined map as the initial map. For phiCPV4, 3D classification was run with no alignment, with the intention of separating out the high-resolution contributing particle images. Five K seeds were used as the basis for running 3D classification in RELION. 3D maps that corresponded to 12-mer complex with sufficient domain-level details were selected and used for further 3D refinement, reducing the angular search to 1.8 degrees and translation shift search to 0.5 pixels. The overall outline of processing is shown in Appendix A. This resulted in refined maps at resolutions of 3.2 Å for GA1 and 2.9 Å for PhiCPV4 (0.143 FSC criterion) (Appendix A). The resulting maps were sharpened using postprocessing to generate maps for atomic model building and fitting. 

### 2.5. Model Building and Refinement

For fitting the cryo-EM density maps, individual monomers were segmented from the sharpened maps using Chimera tools [30,31], and segments corresponding to one subunit were used for the initial model building. 

For the GA1 model, the phi29 gp10 portal protein structure (PDB: 1FOU) was used as the initial model to fit and refine the cryoEM map. First, the phi29 model was rigid body fitted into the cryoEM map of the GA1 portal complex. A single monomer was segmented out of the 12-mer cryoEM, together with the pdb model (phi29 gp10 PDB: 1FOU) and manually refined using Coot [32]. The residues of the phi29 monomer sequence were consecutively replaced with the residues corresponding to the GA1 sequence and fitted into the cryoEM map of GA1 in Coot. The built model was refined in Phenix using the ‘real space refinement’ option. To further improve the model geometry, the monomer model was refined using ISOLDE [33] (molecular-dynamics flexible fitting) and visually inspected in Coot for any rotamer clashes as well as for quality of fit in the cryoEM map. The refined single subunit atomic model for each 12-mer complex was symmetrised using the option in Chimera to generate 12-mer atomic models that were used for interpretation. 

The phiCPV4 gp17 3D map had better resolution, and the model was built using a homology model generated by the I-TASSER server [34]. The best scoring model was used to fit into the 3D EM map in Coot [32]. The process of fitting was initiated by fitting two α-helices of the stem domain built manually and refined in Coot, using the amino acid sequence of the portal monomer. The entire atomic model was then built *de-novo* and completed by adding residues across the monomer chain in Coot, based on the cryoEM map. The resulting model was refined against the EM map for model parameters using the real space refinement option in Phenix [35]. To improve the Ramachandran outliers, ISOLDE was used to refine certain parts of the model backbone as well as clashes in the model [33]. Based on the improved validation score of the model, the model was then symmetrised using Chimera [30,31]. Both models were assessed for their quality using MolProbity [36] and Phenix model validation tools. All figures were prepared using Chimera and Chimera X [30,31].

## 3. Results

### 3.1. Overall Organisation of GA1 gp10 and phiCPV4 gp17 PPs

PPs GA1 gp10 and phiCPV4 gp17 were recombinantly expressed in *E.* coli hosts and purified by column chromatography as described in the methods. Whilst PPs are dodecameric within capsids of fully assembled phages, PPs expressed in the absence of other viral proteins (assembly-naïve conditions) produce C11, C13 and C14 oligomers [27,28]. Both GA1 gp10 and phiCPV4 gp17 form oligomers with varieties in rotational symmetry when purified in assembly-naïve conditions with 30–40% of oligomers having C13 symmetry, some distorted particles and about ~40% exhibiting C12 oligomers. Interestingly, the tags attached to the polypeptide chains were shown to affect the phi29 PP oligomerisation inducing formation of aberrant structures, termed rosettes, that are composed of several portal complexes [37]. For GA1 gp10 the tag had to be proteolytically cleaved to prevent the formation of rosettes and the buffer pH adjusted, to obtain sufficient oligomeric particles for cryo-EM (Materials and Methods Section 2.2; Appendix A). In the case of phiCPV4 gp17, tagged oligomers were obtained via liquid chromatography (see Methods) for subsequent imaging by cryo-EM. As phiCPV4 gp17 rosettes only formed 10% of the observed PP oligomers in the micrographs, the protein was left uncleaved. Particle images of both samples were extracted using RELION.

Both data sets (GA1 gp10 and phiCPV4 gp17) were subjected, in the first steps of processing, to asymmetrical 3D analysis followed by the first round of 3D classification. Particle images that comprised 3D classes with clear C12 symmetry were re-extracted from low-dose frames (See methods). Subsequent 3D refinement of the classes having C12 symmetry in both PPs were performed with imposed C12 symmetry (Figure 3A,B). The PP structures displayed common overall shape with the phi29 portal dodecamer. The GA1 gp10 dodecamer has a height of 78 Å and the outer diameter of 152 Å. The height and diameter the phiCPV4 gp17 dodecamer are 82 Å and 158Å. The diameter of the PP central channel was measured at the narrowest part of the portal protein within the clip domain (Figure 3A,B). In GA1 gp10 this diameter is ~34 Å, and in phiCPV4 gp17 it was similar, at ~36 Å. These sizes are consistent with the diameter of phi29 gp10 channel ~34 Å. The inner diameter sizes of the PP channels of *Salasmaviridae* and *Guelinviridae* phages are slightly bigger compared to the PPs of larger phages of *Caudovirales* described to date [6,27,38] (Table 1). 

The final refinements of GA1 gp10 and phiCPV4 gp17 yielded reconstructions at resolutions of 3.2 Å and 2.9 Å (at a threshold of 0.143) respectively (Figure 3C, Appendix A). The subunit domains such as the wing, stem and clip are readily recognizable in both structures. Both PP maps revealed secondary structure elements showing density details corresponding to amino acids. For GA1 gp10, large residues such as tryptophan and tyrosine were easily identifiable; in the phiCPV4 gp17 map, sidechains were better defined, and small residues were recognisable (Appendix A). The quality of maps allowed *de novo* building (partially for GA1 and for the nearly complete chain of phiCPV4) and refinements of atomic models for both proteins (Appendix A).

In both structures, the α3 and α5 helices of the stem domain have the highest resolution at ~2.5 Å, whilst the clip domain and outer area of the wing domains are at a lower resolution indicating their flexibility (Appendix A). The N-terminus (or ‘armpit’ region) was less defined in the phiCPV4 PP, possibly due to the flexible hexahistidine tag. As neither of the two portal proteins was imaged and reconstructed in the context of DNA, the tunnel loops remained flexible and 3D classification did not reveal clear features of the tunnel loops. 

### 3.2. Crown Domain

The crown domain is formed by the C-terminus of the PP, as in all known Caudovirales phages, but appears highly flexible in many phages as exemplified by phi29 [6,8] and P68 [19]. The crown domains of GA1 gp10 (residues 283–306) and phiCPV4 gp17 (residues 281–302) are short compared to the PP from phages with larger genomes (from 45kb in SPP1 to 41.7 kb in P22) and do not form a defined secondary structure (Figure 3C,D and Appendix A). They are disordered under naïve-assembly conditions having a flexible conformation like in phi29 PP prior to genome ejection [8]. Since these regions of the portal proteins were not resolved, it was impossible to identify any structural differences in these PPs. 

### 3.3. Wing Domain

The wing domain is well defined in both GA1 gp10 and phiCPV4 gp17 PPs, and is formed by two α2 and α6 helices, and a group of peripheral β-strands (Figure 3C,D). The fold of the wing domain is well conserved between GA1 gp10, phiCPV4 gp17 and phi29 gp10. Residues Trp_40_, Glu_53_ and Arg_263_ in GA1 gp10, and Trp_40_, Glu_53_ and Arg_261_ in phiCPV4 gp17 correspond to residues Trp_41_ of β1, Glu_54_ of α2 and Arg_261_ of α6 in the PP of phi29 gp10 (Figure 4A and Appendix A). Helices α2, α6 and strands β1-β6 have similar arrangements in both PPs, accompanied by the interactions between conserved amino acids. They form a core in the wing domain, making it stable and thus essential for the folding of PPs in *Salasmaviridae* and *Guelinviridae* phages. 

The peripheral β-complex in GA1 gp10 includes beta strands that are homologous to the β1-β7, and β11 strands in phi29 gp10 (PDB: 1H5W) although β7 is not clearly defined in the GA1 wing. Helix α2 is located nearly in the middle of the wing domain and tilted 50–60° with respect to the vertical axis of dodecamer. Helix α6 (residues 256–274 in GA gp10 and in phiCPV4 252–271) spans the subunit from the inner rim of the central channel to the outer edge of the wing along the radial direction and forms the upper rim of the wing domain (Figure 3C,D and Figure 4A). Assessment of structural similarity of the wing domains between GA1 gp10 and phi29 gp10 RSMD in aligned atomic models demonstrated that wings in GA1 and phi29 PP have nearly identical folds (Table 2), which is not surprising considering that they have a very high percentage (52.4%) of sequence identity.

In the PP structure of phiCPV4 strands β1 and β11 are readily identified, while the β2–β7 area appears to be less well defined; however, they are similar to the phi29 beta-cluster in the final model (Figure 3, Figure 4A and Appendix A). The β11 strand goes in the opposite direction to the β1 strand and forms a beta layer (Figure 4A and Appendix A) between α2 and α6 within the wing domains in both GA1 gp10 and phiCPV4 gp17. The β11 strand is located on the top of the α6 helix and makes the link with the crown domain. Such an arrangement of these β-strands is also found in phi29 gp10 (PDB code 1H5W) indicating that it is a conserved fold. Helix α6 (residues 252–271) of phiCPV4 gp17 spans the subunit in a well-defined way, as in the GA1 (or phi29) wing (Figure 3C,D). Analysis of the wing domains of phiCPV4 and GA1 with phi29 based on RSMD demonstrated that the phiCPV4 wing domain has more significant differences compared to its phi29 counterpart, whereas the GA1 wing domain, while different from the wing domain in phiCPV4, is quite similar to the phi29 wing domain (Table 2).

### 3.4. Tunnel Loop 

The tunnel loop is the region in the portal protein located between the wing and stem domains (α6 and α5 22 AA) that points to the inner part of the central tunnel. It has been suggested that these loops make contacts with the DNA and together with helices α3 and α5 act as a pumping motor [6]. The inner part of the tunnel loop (four to five residues) is very flexible in both GA1 gp10 and phiCPV4 gp17; the diffused densities in this area did not permit tracing of the polypeptide chain (Figure 3C,D, shown in grey ovals). Interestingly, this part of the chain remains the most conserved region, with residues Glu_233_ to Val_239_ in phi29 gp10 being semi-conserved in both *Salasmaviridae* and *Guelinviridae*. Residues Lys_234_, Lys_235_ and Arg_237_ in phi29 gp10 have been shown by molecular dynamics modelling *in-silico* to closely interact with DNA [39,40]; these residues are conserved in GA1 gp10 and phiCPV4 gp17 (with Lys_235_Arg in phiCPV4 gp17), supporting the evidence that these residues are essential to the portal protein during DNA packaging [41].

### 3.5. Stem Domain

The stem domain comprises two helices: α3 (~20 residues) and α5 (~20 residues), and the N-terminus (~40aa) (Figure 3C,D, shown in orange ovals, and 4B), which is a feature conserved in *Caudovirales* and Herpesvirus PPs, indicating an ancestral fold. These two helices α3 and α5 are found in all PPs described to date [12]. Both GA1 gp10 and phiCPV4 gp17 form dodecamers with most of the inter-monomeric interactions occurring between the stem and clip domains. GA1 gp10 and phiCPV4 gp17 utilise a relatively large number of inter-monomer hydrogen bonds to hold monomers within the dodecameric complex together. In GA1 gp10, four hydrogen bonds are located between the α3 helices of the neighbouring subunits Tyr_139A_, Thr_149A_, Gln_154B_, Lys_147B_ (Figure 4B, bottom panels). Likewise, monomers of phiCPV4 gp17 are held together by 4 hydrogen bonds between the α3 helices: Gln_152A_, Tyr_145A_, Lys_165B_, Asn_158B_, where A and B designate adjacent subunits. Both GA1 gp10 and PhiCPV4 gp17 form dodecamers with most of the inter-monomeric interactions occurring between the stem and clip domains (Figure 4B).

The N-terminus of GA1 gp10 is relatively well defined in the EM map and could be traced from residue Tyr_8_. The N-terminus forms two helical elements on the outer part of the PP complex where the short α0-helix comprises residues Tyr_8_ to Arg_18_ and is positioned in a horizontal orientation with respect to the central axis of the PP. Arg_18_ forms a salt bridge with Glu_148_ of the adjacent subunit (Figure 4B, bottom left and middle panels). This helix is linked through residue Arg_18_ to the α1-helix that spans through the Gly_17_ to Met_37_ residues and is oriented in a nearly vertical direction relative to the PP central axis. 

In phiCPV4 gp17 the N-terminus could be traced from residue Val_23_ and forms the α1 helix similar to that of phi29 gp10 and GA1 gp10 (Figure 4B, bottom right panel). The phiCPV4 N-terminal extremity is not defined, suggesting that as in phi29 gp10 it is also flexible in assembly-naïve conditions. In the procapsid of phi29, the N-terminal end of the PP α1 helix contacts the P-domain of the capsid protein [8]. Comparison of these domains in GA1 gp10 and phiCPV4 gp17 with phi29 gp10 using RSMD assessments reveals high levels of structural preservation in folding of the stems within these phages (Table 2).

### 3.6. Clip Domain

The clip domain is located below the stem domain. The clip spans residues 160–208 in GA1 gp10 and 152–206 in phiCPV4 gp17 between the α3 and α5 helices (Appendix A). The overall organisation of the clip domain in all three phages includes the α4 helix and three β-strands (Figure 5A). However, there are interesting differences in the arrangement of the clip domain between the phages: GA1 gp10 has a beta sheet formed by β8 and β10 strands from one subunit and the third β9 located on the outer edge of the beta sheet is from the adjacent subunit. The interactions (primarily via hydrogen bonds) between adjacent subunits within the clip domain take place between β8 of one subunit and β9 of another. These interactions in GA1 gp10 are very close to those observed in phi29 gp10 having an RSMD of 2.89 Å (Table 2). Increasing of the RSMD is related to small shifts of beta sheet β9 and helix 4 that have slightly different angles (Figure 5B, left panel: overlay of clip domains). Helices α4 do not interact with the adjacent subunits either in phi29 or in GA1 gp10 [8,42]. 

Interestingly, in phiCPV4 gp17 the β-sheet of the clip is formed by three strands β8, β9, and β10 of the same subunit (Figure 5B,C, and Appendix A). In phiCPV4 the inter-monomeric interactions occur between α4 (Asn_176B_, Met_180B_, Leu_183B_, Tyr_184B_) and β9 (Leu_195A_, Val_196A_, Glu_198A_) of the adjacent subunit. Within the dodecamer, β9 of phiCPV4 is homologous to β9 in phi29 gp10. Interactions within the parallel β-strands in phiCPV4 gp17 are intra-monomeric (Figure 5C). The insertion of the Lys-Leu-Ala-Glu-Ala motif (residues 197–202 forming helix α4′) followed by the loop Gln-Ala-Leu-Gln-Glu, makes the link between β9 and β10 bigger, allowing them to form a common β-layer (Appendix A). 

The clip region interacts both with the lower collar protein (gp11) and possibly with the terminal proteins in phi29 [8]. The identification of variation in clip domain organisation helps in our understanding of how phiCPV4 gp17 contacts in the context of the other phage proteins, shedding light on the divergence of the clip domain, and suggests engineering possibilities based on the additional insertions or deletions that modify interactions with tail proteins.

## 4. Discussion

Using cryoEM, we determined the structures of two portal proteins: GA1 gp10, and phiCPV4 gp17, which are phages of the *Salasmaviridae* and *Guelinviridae* families respectively. Both GA1 gp10 and phiCPV4 gp17 PPs have the canonical fold and domain organisation found in the portal protein phi29 gp10 [6,43]. GA1 gp10 has significantly higher identity (52%) with phi29 gp10 compared to the phiCPV4 PP (29%), nonetheless both are found to share the fold of phi29 PP (RMSD 2.18 Å for C_α_). The most interesting differences are observed in the clip domain, which is involved in interactions with tail proteins.

The crown domain of PPs of *Salasmaviridae* (GA1 gp10, phi29 gp10) and *Guelinviridae* phages (phiCPV4 gp17) contacts inner capsid proteins and the packaged DNA genome. There is no clearly definable secondary structure, unlike SPP1 (*Siphoviridae*), P22 (*Podoviridae*), T4 (*Myoviridae*; *Tevenvirinae*; *Tequatrovirus*) and other phages with genomes larger than 40kB, whose crown domains have α-helical folds in the vicinity of the wing domain. Nevertheless, it should be noted that in the PP of the genome emptied phi29 phage, the crown domain (the C-terminus) of the phi29 PP has one α-helix located close to the upper surface of the wing domain [8]. Sizes of the PP polypeptide sequences corresponding to the crown domains in *Salasmaviridae and Guelinviridae* phages appear to be much smaller compared to the crown domain sequence of T7 that forms a helix [43]. The PPs of P22 [44] and SPP1 [10,27] have longer C-termini that fold into a few helices. 

The PP wing domain is the most divergent in *Caudovirales* phages. However, in all phages, it serves as a linker for interactions of the portal protein with the adjacent capsid proteins within the pentameric vertex. Their conformations within the capsid vary due to differences in the interactions between protein components that form 12-fold symmetry in the PP and 5-fold symmetry in the capsid. The wing domain organisation in GA1 gp10 and phiCPV4 gp17 has the classical feature of PPs comprising two helices, α2 and α6. The major spine α6 helix located on the top of the wing is present in all PPs described to date. The main difference between *Salasmaviridae* and *Guelinviridae* phages is that the α6 helix (21 ± 5 aa) is on average shorter by 11 aa compared to large phages, where α6 helix comprises ~32 aa on average. The β1–β7, and a β11 strands of GA1 gp10 and phiCPV4 gp17 form three anti-parallel β-sheets and a β-hairpin formed by two antiparallel beta strands (β1 and β11) on the periphery of the wing (Figure 4A). However, it should also be noted that peripheral beta sheets vary significantly between different phages: the PPs of GA1 gp10, phi29 gp10 [8], and phiCPV4 gp17 have their outer anti-parallel β-strands directed clockwise, when viewed from the side of the crown of the portal protein. PPs of SPP1 [9], T7 [43] and P23-45 [45] have these β-strands arranged in the opposite, counter clockwise, direction. Interestingly, phage T4 does not have such a clear arrangement of the outer hairpin. Instead, in this area the polypeptide chain protrudes outwards as a loop [38]. Apparently, this area of the wing domain is not conserved, and its fold is defined by the specific protein sequence and interactions with the capsid proteins.

The most stable and common feature of all known structures of the PPs, exemplified in the PPs of GA1 gp10, phi29 gp10, and phiCPV4 gp17, is the presence and organisation of the stem domain (α3, α5, and N-terminus). The stem domain has the most conserved fold between the PPs of different phages. In the PPs of GA1 gp10, phi29 gp10, and phiCPV4 gp17, the stem domain comprises two α-helices that form the central channel within the portal dodecamers. Helices α3 and α5 have conserved sequences of similar length, implying that the same interactions between these helices provide inter-subunit interactions that hold the dodecamer together. It is highly likely that these conserved elements of the stem domain indicate an ancestral origin and its role as a mainstay or ‘spine’ for other domain-level variations related to specific interactions and functions such as with the clip domain.

The overall fold of the clip domains in GA1 gp10 and phiCPV4 gp17 is quite similar to the phi29 clip domain: They are composed of helix α4 and two antiparallel β-strands (three in the case of phiCPV4 gp17), located at the bottom of the PP establishing the tail-interacting interface. The inter-monomeric interactions are located between β9 and α4 in phiCPV4 gp17, which is different from all other PPs; while in phi29 and GA1 gp10 it is the β8 and β9 from adjacent subunits that make the more typical inter-monomeric interactions. This is an unexpected observation, since all other *Caudovirales* PPs described to date have the β-sheet of the clip domain formed by strands from adjacent subunits. This variation highlights the subtle ways in which portal architectures can be fine-tuned by nature to perform their functions, relating to specificity. The fact that the clip domain architecture can vary across related portal proteins means that the end of the portal that interacts with the tail apparatus can perform some form of regulatory role, either during tail assembly or during genome gatekeeping (entry or exit). Further structural and functional studies, such as in the context of entire phage, are now required to validate the importance of the new observation observed in the clip domain of phiCPV4 phage PP. 

The larger size (~34Å) of the narrowest diameter in *Salasmaviridae* may be necessary to fit part of the terminal proteins which are covalently attached to the 5′-ends of the dsDNA in phi29 and GA1 [8,46]. Once the genome has been encapsidated, the terminal proteins of phi29 serve as a nucleation point for the collar protein gp11 [8]. Similarly, phiCPV4 gp17 also features a relatively large size (36 Å) at the narrowest point in the PP diameter. Since the related *Guelinviridae* phage phi24R is purported to have a terminal protein [47], it is likely that phiCPV4 also has a similar although so far unidentified terminal protein. The genome of phiCPV4 also contains terminal repeats at the genome ends, which is characteristic of a protein primed replication mechanism and the phiCPV4 polymerase is highly similar to that of phi29, further indicating a common role of terminal proteins in the *Salasmaviridae* and *Guelinviridae* families [48].

## 5. Conclusions

The structures reported in this study and their comparative analysis enable us to consider ways in which chimeric phages can be engineered via modifications to the portal proteins. As stated earlier, a chimeric portal could in principle attach the capsid of one phage to the tail of another. Such chimeragenesis has been demonstrated previously to: (1) generate phage bodies with broader host ranges [49], and (2) combine structural biology insights with synthetic biology to expand the host range of a phage by creating a chimera of its receptor binding protein [1,50]. The last task is a more challenging enterprise: To create a PP chimera *and* to retain its dynamic properties in morphological complex as a genomic gatekeeper. Our new PP structural insights expand the available knowledge of *Caudovirales* portal proteins and will support future studies in PP engineering. 

These new structures of portal proteins further offer a platform on which *Guelinviridae* and *Salasmaviridae* phage chimeras can be designed to infect specific bacteria. Due to their compact nature, small genome size, and their morphological relationship to the archetypal phi29, these phages offer an excellent opportunity for targeted genetic manipulation. By identifying specific areas of interaction with capsid or tail proteins, a phi29-phiCPV4 chimera of the portal protein could be built with a modified wing, stem, or clip domain. This would enable the chimeric phages to deliver modified genetic programs to desired gram positive bacterial species in the available host range. Given that a self-replicating synthetic episome based on the phi29 genome and terminal proteins has already been published [51], it is envisaged that a therapeutically effective genetic sequence could be packaged in the chimeric phage in vitro and delivered to a pathogen in the available host range such as *C. perfringens* [52]. 

GA1 is, like the archetypal phi29, a *Salasmaviridae* phage but targets the pathogen *Bacillus pumilus*, relevant to serious infant infections. The phiCPV4 phage is part of the *Guelinviridae* family and infects *Clostridium perfringens*, a problematic industrial livestock pathogen, which causes foodborne illnesses in humans. Our results indicate the level of structural insight is now sufficiently detailed to inspire engineering designs aimed at creating functional portal protein chimeras. The challenges to design and engineering of a new biologically active complex are not inconsiderable even now, however, yet such efforts are directly relevant to microbial infections of global importance. 

## Figures and Tables

**Figure 1 viruses-13-02532-f001:**
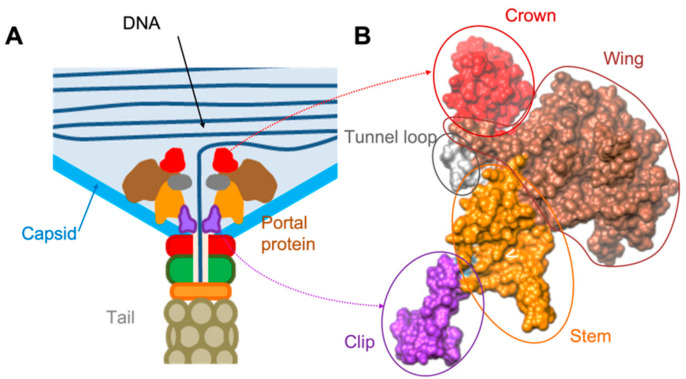
Example of the of the portal protein location within the phage capsid and its domain organisation. (**A**) Overall diagram of the PP position. (**B**) One subunit from the portal protein complex gp6 of the SPP1 phage (pdb 2JES, [10]). Domains are indicated by colouring. Contacts with the capsid are made via wing and stem domains. Clip domains make contacts with accessory proteins of the tail.

**Figure 2 viruses-13-02532-f002:**
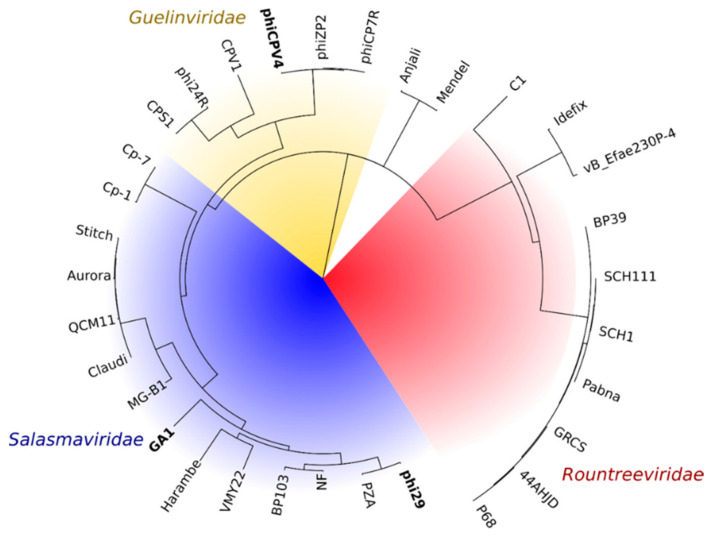
Phylogenetic analysis of related PPs in the order *Caudovirales*. *Rountreeviridae* (infecting *Staphylococcus*, or *Enterococcus* species) are shown in red. *Salasmaviridae* (infecting *Bacillus* species) are shown in blue. *Guelinviridae* (infecting *Clostridium* species) are shown in yellow.

**Figure 3 viruses-13-02532-f003:**
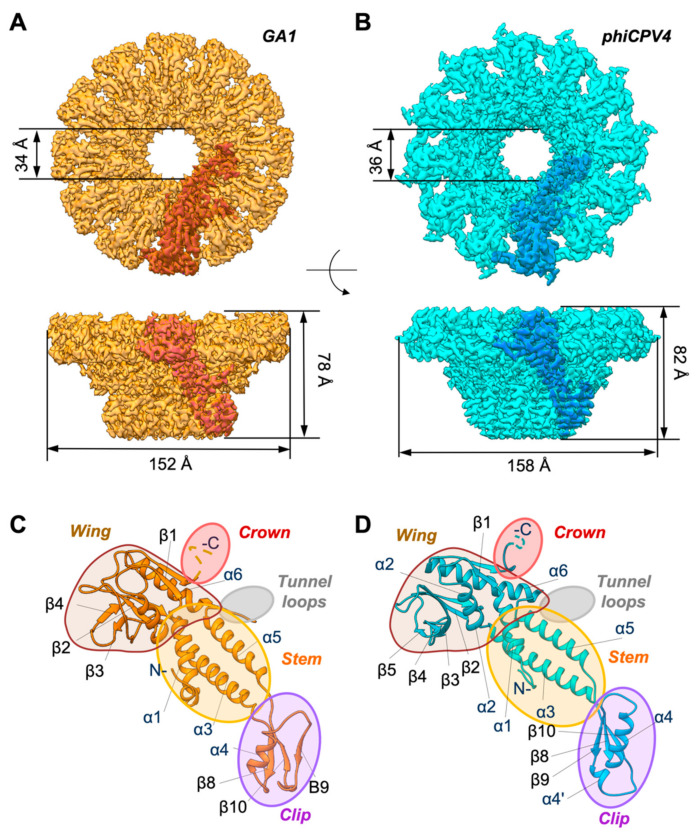
Structures of the GA1 and phiCPV4 portal proteins. Sizes are as indicated; indicative monomer subunits are shown in deeper shade of orange on light orange and blue on cyan, respectively. (**A**) Top and side views of the GA1 PP. (**B**) Top and side views of the phiCPV4 PP. (**C**) Atomic model of the GA1 PP. (**D**) Atomic model of the phiCPV4 PP. Crown domains and DNA loops were not resolved (shown in pink and grey ovals correspondingly). The clip domain of the phiCPV4 PP is slightly bigger due to the presence of the additional short α4′ helix.

**Figure 4 viruses-13-02532-f004:**
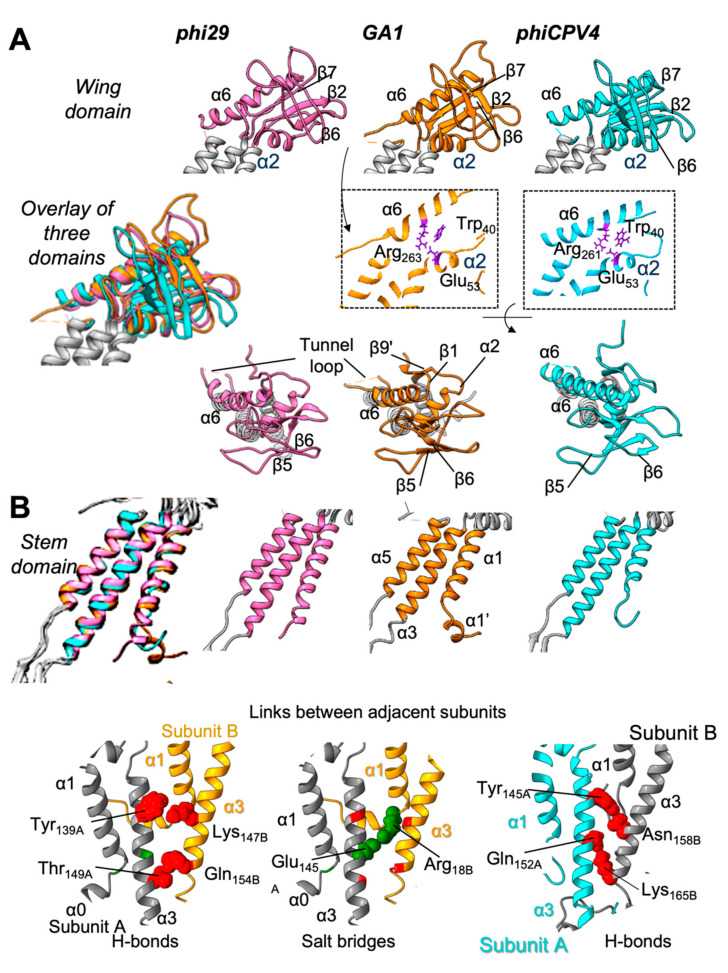
Comparison of the wing and stem domains of the GA1 and phiCPV4 portal proteins. (**A**) Wing domains of phi29 (in pink), GA1(in orange), and phiCPV4 (in cyan). Middle panels show thin slices through helices 6 and 2 indicating interactions within the core of the wings. (**B**) Stem domains of phi29, GA1, and phiCPV4 (colour coding as in **A**), comprising helices α1, α3, and α5. The bottom panel shows interactions between helices of adjacent subunits α3_A_ and α3_B_ in GA1, and in phiCPV4. Hydrogen bonds are shown in red; a salt bridge is indicated in green.

**Figure 5 viruses-13-02532-f005:**
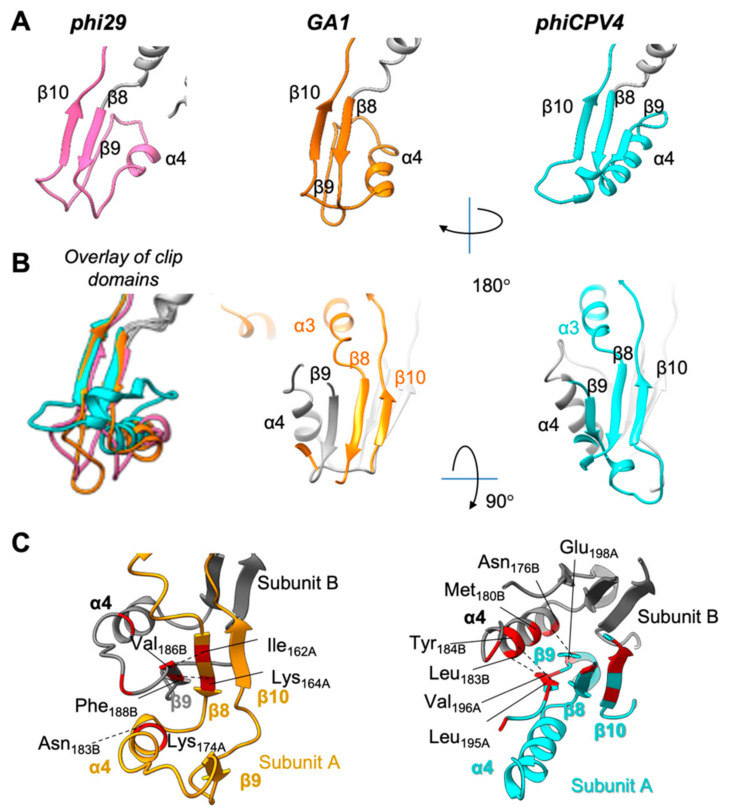
Comparison of the clip domain of the GA1 and phiCPV4 portal proteins. (**A**) Clip domains of phi29, GA1, and phiCPV4 in one subunit (colour coding as in Figure 4). (**B**) Arrangement of β-strands in adjacent subunits. The subunits A are shown in the corresponding colour to the respective PP (GA1, and phiCPV4); the subunits B are shown in grey. (**C**) Interactions between adjacent clip domains of GA1, and phiCPV4: residues forming hydrogen bonds are shown in red. View from the top.

**Table 1 viruses-13-02532-t001:** Overall characteristics of portal proteins from several phages. Inner diameters of the central channels in the portal protein clip domains.

Portal Protein (PP)	Molecular Wight of PP (kDa)	The Length of the Polypeptide Chain (aa)	Narrowest Diameter within Clip Domain (Å)	Genome Size kb	References
phi29 gp10	35.9	309	34	19.2	[6]
GA1 gp10	35.3	306	34	21.2	This MS
phiCPV4 gp17	34.6	301	36	17.9	This MS
SPP1 gp6	57.3	503	30	45	[27]
T4 gp9	61	524	32	168.9	[38]

**Table 2 viruses-13-02532-t002:** Assessment of RSMDs between corresponding domains of phi29, GA1, and phiCPV4 portal proteins.

	GA1 vs. phi29	phiCPV4 vs. phi29	GA1 vs. phiCPV4
Wing domain	1.8	7.3	5.2
Stem domain	0.69	0.93	0.61
Clip domain	2.85	8.4	8.98
Overall	3.06	5.86	5.9

## Data Availability

The GA1 and phiCPV4 maps and the atomic models are deposited in the EMDB database under accession codes EMD-13664, 7PV2, and EMD-13665, 7PV4, respectively.

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
