# Peer review of "Cryo-EM Structures of Two Bacteriophage Portal Proteins Provide Insights for Antimicrobial Phage Engineering"

_viruses, 2021, doi:10.3390/v13122532_

Round 1

Reviewer 1 Report

The manuscript entitled “A comparative analysis of portal proteins of GA1 and phiCPV4 phages targeting gram-positive pathogenic species using cryo-EM” by Javed, et. al reports two phage portal protein (PP) structures determined by cryoEM.  Through structural comparison of the two structures with that of a phi29 homolog, they found that the clip domain in phiCPV4 PP displays a different packing. The results may be of interest to researchers working on phage structures and phage engineering to develop alternative strategies against antibiotic resistance.  The cryoEM technique is well presented. The manuscript is overall well written but contains too many details on the description of the two structures. Below are comments to help improve the manuscript before it may be accepted for publication.

  1. Title: The authors report the analysis on PP structures from three phages (GA1, phiCPV4, and phi29). The phi29 PP structure was solved 20 years ago. There is no novelty here in the title to emphasize on “targeting gram-positive pathogenic species” as phi29 also targets gram-positive pathogens. It is thus suggested to have a revised title to better summarize the work (i.e, two cryoEM structures).
  2. Abstract: The authors claim the need of engineering phages for killing gram-positive bacterial pathogens. However, there is an inadequate background introduction of the current progress on structures and phage engineering toward this goal. The authors should add 1-2 sentences to justify the need of structural work toward phage engineering.
  3. Figure 1: Is panel B a theoretical model or an experimental structure? Please add its pdb code and reference if the structure is known.
  4. Figure 2: In addition to using colors to code the three clades, it’s better to label their clade names directly on the figure. The figure caption should be in lower case except the initial letter.
  5. Materials & Methods: p4, lines 123-125. Presumably, this part is about the cloning work for phiCPV4. It should then be in the same paragraph as cloning GA1. What primers were used for cloning phiCPV4 pp gene?
  6. Materials & Methods: p4, lines 132-136. This part is about bioinformatics and should be separated from 2.1. Cloning, expression and purification. A new section is needed.
  7. Page 5, line 145. Remove “per frame” which is redundant to the text on the following line.
  8. Materials & Methods: p5, lines 153-161. The authors wrote that “Particles were picked using a combination of manual and automated picking in RELION.” But then they wrote that “… as templates for auto-picking using Gautomatch”. The authors should revise their text to make it clear what program was used for what.  
  9. P6, line 198 and line 208. Please use “real space refine” in both places.
  10. Figure 3: The figure quality is poor; distracting gray lines (out of context) can be clearly seen. In Fig. 3C, the crown domain is disordered and should be drawn as a dashed line, instead of nothing there. The same to the tunnel loop(s) in Fig. 3D.
  11. P8, line 238. Please use “width” instead of “diameter”.
  12. Figure 4: Similar to Fig. 3, its image quality is poor. Based on the superimposition, the three structures are similar for the wing and stem domains. The authors should make the superimposition figure large enough to see details (equivalent to these details for individual structures). As mentioned in my summary report, there are too many details on the description of structures in sections 3.3 and 3.5. The authors should simplify their text in both sections.  Missing RMSD values for the superimposed overall structures as well as for individual domains.  
  13. P10-11: lines 318, 320, and 326. The authors should include plots of densities (as supplemental materials) for those side chains involved in hydrogen bond interactions or salt bridges. Otherwise, there is no way to justify the formation of hydrogen bonds and the salt bridge.  
  14. PDBs and density maps need to be deposited to their respective databases.
  15. Figure 5: The same as Figs. 3 and 4, its image quality is poor. In addition, the bottom two panels in Fig. 5B are essentially duplicated with panel C and may be removed. In panel C caption, what does the red color mean?
  16. The authors propose to provide structural details/knowledge for engineering designs. As the whole structure is known for phage phi29 (ref 7), can authors superimpose their structures with the tail structure of phi29 and offer some insights (in discussion) on phage designing such as tail switching?  

Author Response

We thank the reviewers for the helpful comments and good suggestions related particularly to amendment of the figures, which helped us to improve the manuscript. The comments of reviewers are shown in blue font (or italic); our responses are in black font.

Answers to reviewer 1. 

Title:  We have considered this suggestion and have modified the title as follows: Cryo-EM structures of two bacteriophage portal proteins provide insights for antimicrobial phage engineering”.

Abstract: The authors claim the need of engineering phages for killing gram-positive bacterial pathogens. However, there is an inadequate background introduction of the current progress on structures and phage engineering toward this goal. The authors should add 1-2 sentences to justify the need of structural work toward phage engineering.

We have added a new sentence spanning lines 37-40 in the Introduction; since the abstract limits the number of words and explains the overall ideas rather than finer details.

Figure 1: Is panel B a theoretical model or an experimental structure? Please add its pdb code and reference if the structure is known.

That has been corrected. This was a model structure of the monomer from the 13-mer of the spSPP1 portal protein. The citation was added.

Figure 2: In addition to using colors to code the three clades, it’s better to label their clade names directly on the figure. The figure caption should be in lower case except the initial letter.

The figure has been corrected.

Materials & Methods: p4, lines 123-125. Presumably, this part is about the cloning work for phiCPV4. It should then be in the same paragraph as cloning GA1. What primers were used for cloning phiCPV4 pp gene?

The information has been added.

Materials & Methods: p4, lines 132-136. This part is about bioinformatics and should be separated from 2.1. Cloning, expression and purification. A new section is needed.

Thank you for this advice. That part has been modified and the new subsection has been added.

Page 5, line 145. Remove “per frame” which is redundant to the text on the following line.

We thank the reviewer for spotting this linguistic inaccuracy. That has been corrected.

Materials & Methods: p5, lines 153-161. The authors wrote that “Particles were picked using a combination of manual and automated picking in RELION.” But then they wrote that “… as templates for auto-picking using Gautomatch”. The authors should revise their text to make it clear what program was used for what.

This part of the Methods was corrected and extended. The procedure has more details now. We hope that the text is more coherent now.

P6, line 198 and line 208. Please use “real space refine” in both places.

That was done.

Figure 3: The figure quality is poor; distracting gray lines (out of context) can be clearly seen. In Fig. 3C, the crown domain is disordered and should be drawn as a dashed line, instead of nothing there. The same to the tunnel loop(s) in Fig. 3D.

We are sorry that the reviewer has got an impression that the figures were of poor quality. Something clearly went wrong during the additional compression of the file, when it was prepared for the reviewers since in our PDF file, we did not see this problem.

We have tried to improve the figure and have drawn the possible areas of the crown domain as a dashed line. However, since the domain was not resolved it was non-realistic to provide the more accurate outline of the domain, so we have kept the general oval providing the information where the domain is located.

P8, line 238. Please use “width” instead of “diameter”.

We have changed it for “diameter”, which fits better to the context of the MS section.

Figure 4: Similar to Fig. 3, its image quality is poor. Based on the superimposition, the three structures are similar for the wing and stem domains. The authors should make the superimposition figure large enough to see details (equivalent to these details for individual structures).

The figure has been modified according to the reviewer’s request.

As mentioned in my summary report, there are too many details on the description of structures in sections 3.3 and 3.5. The authors should simplify their text in both sections.

We must politely disagree here with the reviewer; it was difficult to find what has to be shortened. The information was essential in the context of overall differences between these portal proteins.

Missing RMSD values for the superimposed overall structures as well as for individual domains.

We thank the reviewer for this suggestion and have done these calculations (now included in the manuscript). The RSMD provides quite important information referred to in the discussion part of the MS.

P10-11: lines 318, 320, and 326. The authors should include plots of densities (as supplemental materials) for those side chains involved in hydrogen bond interactions or salt bridges. Otherwise, there is no way to justify the formation of hydrogen bonds and the salt bridge.

It was not quite clear what the reviewer wanted to see. The quality of the map has been shown in the supplementary materials with the atomic models fitted into EM maps. We are very sorry, that the reviewers were not able to see the supplementary materials, and this caused additional questions related to the quality of the maps provided.    

PDBs and density maps need to be deposited to their respective databases.

The information related to deposition of data and PDB access codes is provided now.

Figure 5: The same as Figs. 3 and 4, its image quality is poor.

This was related to the poor conversion of the MS into a PDF file, not showing the original quality of the figures.

In addition, the bottom two panels in Fig. 5B are essentially duplicated with panel C and may be removed. In panel C caption, what does the red color mean?

We thank the reviewer for the good advice. The figure has been amended accordingly and the figure legend extended.

The authors propose to provide structural details/knowledge for engineering designs. As the whole structure is known for phage phi29 (ref 7), can authors superimpose their structures with the tail structure of phi29 and offer some insights (in discussion) on phage designing such as tail switching?

The structure of the portal proteins cannot be superimposed to the tail structures, which were not well defined even for Phi29 (they were obtained from the reconstruction of the fully assembled phage) and the resolution of the elements within this tail map was not very high, preventing us seeing details of interaction. We, likewise, would welcome new data allowing a more accurate analysis of such interactions.  

Reviewer 2 Report

The manuscript « A comparative analysis of portal proteins of GA1 and phiCPV4 phages targeting gram-positive pathogenic species using cryo-EM » presents the structure of two dodecameric portal complexes from Salasmaviridae phage GA1, which targets Bacillus pumilus, and Guelinviridae phage phiCPV4 that infects Clostridium perfringens. The cryo-EM maps were resolved by single particle single particle analysis at 3.3 Å and 2.9 Å respectively, thus, a sufficient resolution to built an atomic model. As expected, the canonical fold and domain organization of GA1, phiCPV4 and phi29, the most studied phage of the Salasmaviridae family. This work expand the available knowledge of Caudavirales portal protein, which is fundamental in order to support future studies in portal protein engineering.

Overall, this work is well adapted for publication in Viruses, but some minor points need to be improved before publication :

  1. In « Grid preparation and data collection », it would be interesting to add the blot time and blot force. The number of frames per movie is not indicated.

  1. In « Image processing ». The initial number of particles is not mentioned. Some steps are not clear concerning the initial model used and if or not the C12 symmetry has been applied. Line171, « Both datasets ware ???? », I suppose « were ». For example, in this step, what was the reference model ?

Line 174, « complexes were extracted . The extracted images » ….. Reformulate this sentence. You need to clarify : either you talk about particles or particle images, but not image.

A flowchart would be really helpful. I guess there is one as indicated in the text, but it was present within the supplementary figures file !

In the same way, the map resolution for GA1 and phiCPV4 are missing.

-The assessment of variations in rotational symmetry is not clear. Do you estimate it visually or do impose a Cn symmetry and compute a coefficient correlation with the initial image ??? This point needs to be clarified.

  1. « Model building and refinement ». I don ‘t understand «  The resulting model was symmetrised using Chimera » ? What do you mean exactly here ? The symmetry was not applied in Relion ??? Clarify this point.

  1. In the first part of results, it is mentioned that the cleavage of the tag for GA1 gp10 was required. It is not clear.

  1. Figure 3. It would be better to have GA1 and phiCPV4 in the same orientation (C&D). The wing domain has to be also indicated.

  1. Check all references to figures. It is often written « Figures 3c, d and 4a » …. But in the corresponding figure, it is A,B etc …..

  1. When you start the description of GA1 gp10 and phiCPV4 pg17, it would be interesting to specify which parts of the protein are not resolved in the EM-map. A figure with GA1 gp10 and phiCPV4 pg17aa sequences indicating exactly the position of domains, and the regions resolved would be really helpful for the reader.

  1. The RMSD per domain with phi29 would be also helpful.

  1. Line254 « maps allowed de novo building ». When using phi29 as starting model or I-Tasser …it is not a de novo building, it is a building by homology.

  1. Line 265. What is the total number of residues in GA1 and phiCPV4 ? it is indicated for the crown domain 306aa and 302aa respectively. In the table, protein residues are indicated as 257 ans 256 ???? The molecular mass of the two PP should also be indicated somewhere in the manuscript.

  1. In Figure 4, as GAI1 and phiCPV4  are compared to phi29, the interactions within the core of phi29 are also required, the inter-domain interactions of phi29 also. The last panel needs to be improved. We see squares, and it is not very nice. Moreover, GA1 appears yellow and not orange ! For phiCPV4, the subunit in blue must be inversed to respect the others panels.

12. In Figure 5, phi29 is missing in the B panel. 

Author Response

We thank the reviewers for the helpful comments and good suggestions related particularly to amendment of the figures, which helped us to improve the manuscript. The comments of reviewers are shown in blue font (or italic); our responses are in black font.

Reviewer 2

In « Grid preparation and data collection », it would be interesting to add the blot time and blot force. The number of frames per movie is not indicated.

This information has been updated in Methods.

In « Image processing ». The initial number of particles is not mentioned. Some steps are not clear concerning the initial model used and if or not the C12 symmetry has been applied

This information has now been provided in Methods and shown in supplementary materials figure 2.

Line171, « Both datasets ware ???? », I suppose « were ». For example, in this step, what was the reference model ?

We thank the reviewers for noticing this typo, which has been corrected.

Line 174, « complexes were extracted . The extracted images » ….. Reformulate this sentence. You need to clarify : either you talk about particles or particle images, but not image.

We are agree completely with the reviewer. The sentence has been rephrased.

A flowchart would be really helpful. I guess there is one as indicated in the text, but it was present within the supplementary figures file !

Our apologies to both reviewers. We do not know why the supplementary figures were not provided to the reviewers, since they had also been submitted to the journal.

In the same way, the map resolution for GA1 and phiCPV4 are missing.

This information is given in supplementary materials: we are sorry that the reviewers were not able to see it.

The assessment of variations in rotational symmetry is not clear. Do you estimate it visually or do impose a Cn symmetry and compute a coefficient correlation with the initial image ??? This point needs to be clarified.

We have done the symmetry assessment by the rotational correlation of the classes using Imagic software. However, since it is a well-established fact that in capsids, portal proteins have 12-fold symmetry we did not pay much attention to other structures. Thus, we have selected only images that correspond to 12-fold symmetry and did not consider that analysis of the rotational symmetry as a major result of the research. Selection of the images of particles with 12-fold symmetry was based on two steps: 2D classification and evaluation of rotational symmetry of 2D classes using rotational correlation function and 3D classification. During the first step we assessed percentage of particles that had 12-fold symmetry. The images representing side views were used for the asymmetrical reconstruction (symmetry C1) with the particle images comprising the 12-fold symmetry end view classes. These structures were subjected to 3D classification and the particle images corresponding to side views and utilised in  the structure with the symmetry where used during the refinement with the imposed 12-fold symmetry. That is now clarified in Methods.

« Model building and refinement ». I don ‘t understand «  The resulting model was symmetrised using Chimera » ? What do you mean exactly here ? The symmetry was not applied in Relion ??? Clarify this point.

We clarified this point. The tracing of the atomic model was done only for one subunit based on the cryoEM densities corresponding to one subunit. This model was symmetrised in Chimera and areas of the contacts between subunit were refined in Coot and Phenix.

In the first part of results, it is mentioned that the cleavage of the tag for GA1 gp10 was required. It is not clear.

Tag cleavage was shown to affect the phi29 PP oligomerisation by preventing the formation of aberrant complexes of portal oligomers: the naïve-assembled portal oligomers were forming “rosettes” from 4 or 5 oligomers. In the case of phiCPV4 gp17, tagged oligomers were obtained via liquid chromatography (see Methods) for subsequent use in cryo-EM. In the case of GA1 gp10, the tag initially had to be proteolytically cleaved to prevent the formation of rosettes and the buffer pH adjusted, to obtain sufficient oligomeric particles for cryo-EM (Materials and Methods section 2.2; Figures S2, S3 and S4; Table S1).

Figure 3. It would be better to have GA1 and phiCPV4 in the same orientation (C&D). The wing domain has to be also indicated.

The figure has been amended according to the reviewer’s request.

Check all references to figures. It is often written « Figures 3c, d and 4a » …. But in the corresponding figure, it is A,B etc …..

We thank the reviewer for noticing these omissions. We believe we have now made the requested corrections.

When you start the description of GA1 gp10 and phiCPV4 pg17, it would be interesting to specify which parts of the protein are not resolved in the EM-map. A figure with GA1 gp10 and phiCPV4 pg17aa sequences indicating exactly the position of domains, and the regions resolved would be really helpful for the reader. 

This information has been provided both in the MS text (results) and in supplementary info, which was unfortunately not given to the reviewers, although it was submitted to the journal. We apologise for that. The areas that were not resolved are indicated in the Figure 3.

The RMSD per domain with phi29 would be also helpful.

We thank the reviewer for a good suggestion; this was done, and implemented in the MS.

Line254 « maps allowed de novo building ». When using phi29 as starting model or I-Tasser …it is not a de novo building, it is a building by homology.

Possibly it was a good point, but homology fitting does not correspond exactly to the rigid fitting followed by automated refinement. Certain areas were fitted from scratch, since they did not fit to the densities, as it was with the clip domain.

Line 265. What is the total number of residues in GA1 and phiCPV4 ? it is indicated for the crown domain 306aa and 302aa respectively. In the table, protein residues are indicated as 257 ans 256 ????

The crown domains of GA1 gp10 (residues 283-306) and phiCPV4 gp17 (residues 281-302) are short compared to the PP from phages with larger genomes (from 45aa in SPP1 to 137aa in P22) and do not form a defined secondary structure (Figures 3C, D, S1). They are disordered under naïve-assembly conditions having a flexible conformation similar to the PP in phi29 prior to genome ejection [7].

The text has been amended accordingly.

In Figure 4, as GAI1 and phiCPV4  are compared to phi29, the interactions within the core of phi29 are also required, the inter-domain interactions of phi29 also. The last panel needs to be improved. We see squares, and it is not very nice. Moreover, GA1 appears yellow and not orange ! For phiCPV4, the subunit in blue must be inversed to respect the others panels.

We apologize for the bad quality of the figures, but it was not seen in our version of the file. Something has happened during preparation of files for the reviewers in the editorial office. If the reviewer calls this colour as yellow, this is fine with us and possibly related to the changes in colours at changing formats of the figures. Sorry about that. The last sentence of this comment is a bit confusing.

In Figure 5, phi29 is missing in the B panel. 

In this figure 5B phi29 is shown in the overlay of all three clip domains. The phi29 clip domain coincides well with GA1. Unfortunately, the pink colour was not very well seen.